# Transcriptomic and Metabolomic Analysis Reveal Possible Molecular Mechanisms Regulating Tea Plant Growth Elicited by Chitosan Oligosaccharide

**DOI:** 10.3390/ijms23105469

**Published:** 2022-05-13

**Authors:** Dezhong Ji, Lina Ou, Xiaoli Ren, Xiuju Yang, Yanni Tan, Xia Zhou, Linhong Jin

**Affiliations:** 1State Key Laboratory Breeding Base of Green Pesticide and Agricultural Bioengineering, Key Laboratory of Green Pesticide and Agricultural Bioengineering, Ministry of Education, Guizhou University, Guiyang 550025, China; gs.dzji19@gzu.edu.cn (D.J.); gs.lnou17@gzu.edu.cn (L.O.); gs.xlren20@gzu.edu.cn (X.R.); gs.xjyang20@gzu.edu.cn (X.Y.); gs.yntan20@gzu.edu.cn (Y.T.); 2College of Tea, Guizhou University, Guiyang 550025, China

**Keywords:** chitosan oligosaccharide, tea plant growth and development, transcriptome, metabolic pathways, metabolism

## Abstract

Chitosan oligosaccharide (COS) plays an important role in the growth and development of tea plants. However, responses in tea plants trigged by COS have not been thoroughly investigated. In this study, we integrated transcriptomics and metabolomics analysis to understand the mechanisms of chitosan-induced tea quality improvement and growth promotion. The combined analysis revealed an obvious link between the flourishing development of the tea plant and the presence of COS. It obviously regulated the growth and development of the tea and the metabolomic process. The chlorophyll, soluble sugar, and amino acid content in the tea leaves was increased. The phytohormones, carbohydrates, and amino acid levels were zoomed-in in both transcript and metabolomics analyses compared to the control. The expression of the genes related to phytohormones transduction, carbon fixation, and amino acid metabolism during the growth and development of tea plants were significantly upregulated. Our findings indicated that alerted transcriptomic and metabolic responses occurring with the application of COS could cause efficiency in substrates in pivotal pathways and hence, elicited plant growth.

## 1. Introduction

The quantity and quality of tea leaves are mainly affected by the metabolites in tea leaves, such as tea polyphenols, theanine, caffeine, vitamins, volatile oils, polysaccharides, and minerals, which are closely related to the growth process of tea trees [1]. Tea plant pests and diseases and environmental pressure cause major economic losses for tea. Traditional pesticides can increase the yield of tea, but their residues would be harmful to the local ecosystem and the quality of the tea plant [2,3,4,5,6]. The application of natural-origin chitosan has been reported safe for the environment and beneficial to the quality and yield of crops [7,8,9,10].

Chitosan oligosaccharides (COS) are derived from the shells of shrimp and other sea crustaceans and possess good water solubility and a wide range of biological activity [11]. Chitosan has generated great attention in a range of fields due to its exceptional biological activities, especially in agricultural uses like antimicrobial action [12,13], plant growth and development promotion [14,15,16], plant defenses elicitation [13], and resistance against various abiotic stresses [12,14,17]. For example, it augments plant height, leaf number, fruit weight, number, and yield [18,19,20]. Chitosan can improve the growth shoot and root length in beans [21]. The promotion function of chitosan application is also verified in several other plants, such as potatoes [22], wheat [23], and rice [24]. Chitosan can increase chlorophyll content and thus photosynthesis in rape seedlings [25] or up-regulate a series of primary C and N metabolic pathways in the leaves of wheat seedlings [26]. We have previously reported that chitosan has obvious promoting effects on tea plant growth [27] and eliminates the cold stress on the tea plant [28]. This biodegradable, economical, and renewable oligosaccharide is more appealing in organic agriculture and benefits plant yields and ecological biodiversity.

Most previous studies have been limited to the apparent effects of chitosan on plant physiology and growth characteristics. However, no research has confirmed the metabolic responses of chitosan on tea plant growth and the yield promotion of chitosan in the tea tree. At the same time, plant histological techniques are widely used for the discovery of functional genes, identification of metabolic pathways, and modification of plant cell characteristics [29,30,31]. Metabolite profiling technology is critical to the comprehensive analysis of the plant growth mechanism [32,33,34]. Multiparametric metabolic response of plants can be achieved by metabolomics, where metabolic accumulation and change could be detected by chromatography interfaced with mass spectrometry [32].

However, despite the elicitors function of chitosan discovered to date, alterations or responses in tea plants trigged by it have not been thoroughly investigated. Therefore, biochemical analysis of the tea plant by integrating transcriptomics and metabolomics analysis will aid our understanding of the mechanisms of chitosan-induced tea quality improvement and growth-promoting.

In this study, we compared metabolic profiles of control and chitosan-treated leaves and relationships within the metabolic network of the tea plant. Using this approach, we determined that a chitosan-induced metabolic strategy can increase the normal regulation of amino and nucleotide sugar metabolism, which enables intrinsic growth promotion.

## 2. Results

### 2.1. Yield Measurement of Tea Leaves

The data of tea yield, including bud density, were measured as shown in Table 1. The concentration of COS at 2.0 g/667 m^2^ (1:800) performed better but with no significant difference. The average bud densities were 122.3, 128.8, and126.9 buds for the three concentrations 1.6, 2.0, and 3.2 g/mu (1 mu = 667 m^2^), which increased by 18.97, 25.29, and 23.44% when compared to CK. Accordingly, the mean 100-bud weight was 7.68, 7.77, and 7.70 g and increased by 13.66, 15.06, and 14.03% compared to CK. Actual tea leave yield was 33.96, 36.33, and 34.66 g/m^2^ and increased by 15.7, 23.8, and 18.1%, respectively. In sum, the presence of COS can significantly increase the tea plant yield. In contrast, the concentration of COS at 2 g/667 m^2^ performed better, though with no significant difference.

Among the three tested concentrations, 800-time dilution performed best. Both the higher and lower concentrations spoiled the effect rather than increasing the production yield. Therefore, we applied and checked the most effective 800-time for their function.

As shown in Figure 1, more fresh shoots from the tea bushes were exposed in the COS treatment than in that of the control group. In addition, the promoted emergence of new and tender leaves could be compared by bud density and 100-bud weight.

### 2.2. Changes of Chlorophyll, Soluble Sugar, and Amino Acid Content in Tea Leaves

The effects of COS treatment on the contents of the total amino acids, chlorophyll, and soluble sugar in the tea leaves were investigated. The values for those tested parameters in the COS group were obviously higher than in the control group (Figure 2A). Among them, the amino acids in the COS-treatment group were 60.77 mg/g FW and 39.86% higher than in the CK group (43.45 mg/g FW). Similarly, the chlorophyll content in the COS treatment was 0.41 mg/g FW and 64% higher than in the control (0.18 mg/g FW), with great significance (Figure 2B). In addition, the soluble sugar content of COS treatment was 83.37 mg/g FW and increased by 17.76% compared to the control group (Figure 2C).

### 2.3. Transcriptomic Analysis

#### 2.3.1. Transcriptome Sequencing and Assembly

In this study, both COS and CK group samples were sequenced by the Illumina Nova seq platform. A total of 70–106 million raw reads were obtained for the COS group three repeats and 76 to 96 million raw reads for the CK group, respectively. After filtering, 66.2–100.0 million clean reads were obtained for the COS group and 70–90 million reads for the control. In all groups, Q20 was 100%, Q30 was above 99%, and the average GC contents of the CK group and COS group were also above 46%. The results indicated that the sequencing is of high quality and could be used for further analysis (Table 2).

#### 2.3.2. Differentially Expressed Gene (DEG) Analysis

The biological repeatability between samples was analyzed using Spearman analysis to verify the high correlation of gene expression levels between samples and proved that a perfect Spearman correlation occurred based on the RPKM (Reads per Kilobase per Million) of different samples. Genes with *p*-value < 0.05 and |log2 (Fold Change)| > 1 were defined as DEGs (genes that were differentially expressed between control and COS). A total of 5806 DEGs were identified between two groups of tea leave samples, of which 3262 up-regulated genes and 2544 down-regulated genes were found (Appendix A).

#### 2.3.3. Gene Ontology (GO) Annotation

Gene ontology (GO) provides the representation of gene product attributes and covers three domains: cellular component (CC), molecular function (MF), and biological process (BP). The GO annotation for the two groups revealed that a major part of DEGs was involved in BP, and MF covered DEGs with high rich factors (Figure 3 and Appendix A). The main CC categories (the parts of a cell or its extracellular environment) were “intracellular organelles” (GO:0043229), “cytoplasm” (GO:0005737)”, and “organelles part” (GO:0044422), and so on. The main MF category (the elemental activities of a gene product at the molecular level) was the expression of genes related to transferase activity at the molecular level, such as “methyltransferase activity” (GO:0008168), “transferase activity” (GO:0008168), “serine hydrolase activity” (GO:0017171), and “ubiquitin-protein transferase activity”(GO:0004842). In terms of BP, the main categories focus on metabolic processes or molecular events, such as the “protein metabolic process” (GO:0019538), the phosphorus metabolic process (GO:0006793), and the cellular amino acid metabolic process (GO:0006520).

#### 2.3.4. Kyoto Encyclopedia of Genes and Genomes (KEGG) Annotation

The degree of enrichment of KEGG was measured by the abundance factor, *p*-value, and the number of genes in the pathway. The significance of enrichment is shown on the horizontal coordinates; the greater the value (−log10 (*p*-value)), the more significant the enrichment. The KEGG pathway is shown on the longitudinal coordinates in Figure 4. The size of the dots indicates the number of different genes contained in the KEGG pathway, and the color of the dots indicates the degree of rich factor enrichment. These enriched pathways include “starch and sucrose metabolism” (ko00500), “glycolysis/gluconeogenesis” (ko00010), “biosynthesis of amino acids” (ko01230), “plant hormone signaling” (ko04075) (Figure 4, Appendix A). The DEGs were significantly enriched in pathways for “starch and sucrose metabolism”, “phytohormone signal transduction”,“Phenylpropanoid biosynthesis”, “glycolysis/gluconeogenesis”, “glycerophospholipid metabolism”, “fatty acid metabolism”, “cysteine and methionine metabolism”, “the carbon fixation of photosynthetic organisms”, “the biosynthesis of amino acids”, “the biosynthesis of unsaturated fatty acids”, and other genes closely related to plant growth and development, which were all significantly up-regulated.

The differentially expressed genes (DEGs) involved in response to COS were summed up and analyzed. Many DEGs are found related to pathways related to the growth and development of tea plants, including plant hormone signal transduction, starch, and sucrose metabolism, phenylpropanoid biosynthesis, glycolysis/gluconeogenesis, and carbon fixation in photosynthetic organizations, and biosynthesis of amino acids. The DEGs involved in the main metabolic pathways, including plant hormone signal transduction, carbohydrate metabolism, photosynthesis, amino acid metabolism, and other physiological and biochemical processes related to tea growth and development, are displayed in Appendix A. These DEGs were characterized as genes encoding auxin-responsive GH3 gene family (GH3), SAUR family protein (SAUR), auxin influx carrier (AUX1), histidine-containing phosphotransfer protein (AHP), alpha-amylase (AMY), sucrose synthase (SUS), fructose-1,6-bisphosphatase I (FBP), trehalose-phosphatase (otsB), sucrose phosphate synthase (SPS), sucrose synthase (SS), malate dehydrogenase (MDH), carbonic anhydrase (CA), glutamine synthetase (GLUL), glutamate dehydrogenase(GDH), tryptophan (trpB) synthase and so on. Among them, glutamine synthetase (GLUL) and tryptophan (trpB) synthase are DEGs specifically belonging to tea in biosynthesis amino acids.

#### 2.3.5. Validation of qRT-PCR (Quantitative Real-Time Polymerase Chain Reaction)

To validate the accuracy and repeatability of the transcriptome analysis, qRT-PCR (Figure 5) was performed on a set of DEGs selected randomly from the above acquired DEGs. Results showed that the qRT-PCR expression profiles were consistent with their abundance changes identified by RNA-seq (Appendix A), which verified the reproducibility and credibility of RNA-seq data. Results confirmed that AMY, SUS, SAUR, and GH3 were significantly upregulated by exogenous COS.

### 2.4. Metabolome Data Analysis

#### 2.4.1. Effect of COS on Differential Changes in Tea Metabolites

To examine metabolite patterns elicited by COS, metabolites were extracted from the leaves of the control and COS groups, followed by detection, respectively. Metabolomic analysis of the tea samples by UPLC-QTOF MS in positive mode (ESI^+^) and negative mode (ESI^−^) mode detected. Principal components analysis (PCA) score plots (*n* = 6 for each control and COS-treated sample from tea plant)) revealed discriminative metabolic levels between the tea plant sample of COS and the control group in both pos and neg-modes (Figure 6). The score scatter plots classified samples as COS treatment “HAI7_2” and control “KB7_1”, with 47.57 and 55.91% of the variability contained in PC1 in positive and negative mode, respectively.

By setting the threshold for significantly differential metabolites (DMs) screening at variable importance in the projection (VIP) > 1.0, *p* < 0.05, a total of 416 DMs were obtained. The content of each metabolite was leveled to complete hierarchical linkage clustering as presented in the heatmap. Each sample was visualized in a single column (COS treatment sample: H7_2_1-H7_2_6; control: K7_1_1-K7_1_6), and each metabolite is represented by a single row [Figure 7, Appendix A. Of which 108 DMs were significantly upregulated and 116 DMs were significantly downregulated in positive mode, 90 were upregulated, and 112 were downregulated in negative mode (Table 3).

#### 2.4.2. KEGG Analysis of DMs Response to COS Treatment

KEGG pathways were organized and analyzed to explore the functions of COS on the DMs related to tea plant development (Figure 8). The color of the point represents the *p*-value of the hypergeometric test. The smaller the value, the more reliable and statistically significant the result is. The size of the dots represents the number of differential metabolites in the corresponding pathway; the larger size indicates a higher significant differential of metabolites in the pathway. It could be seen that the DMs were mainly enriched in the following six pathways, “biosynthesis of secondary metabolites”, “Phenylpropanoid biosynthesis”, “flavone and flavonol biosynthesis”, “alpha−Linolenic acid metabolism”, “Biosynthesis of unsaturated fatty acids”, “Arachidonic acid metabolism”, and “Glutathione metabolism” (Figure 8, Appendix A). DMs enriched in these metabolic pathways may play an immediate physiological role and promote and influence growth and development. Among them, the most enrichment of differential metabolites pathway is the biosynthesis of secondary metabolites, and 33 DMs were found in it (Appendix A), including amino acids (L-Histidine, L-Ornithine, L-Asparagine), organic acids (Loganic acid, Ferulic acid, Ascorbic acid), flavonoids (Rutin, Kaempferol, Quercetin), and other compounds. The above results showed that COS treatment could significantly increase the production of secondary metabolites in tea plants, thereby promoting the growth and development of tea plants.

### 2.5. Combination of the Metabolic Profiling and Transcriptomic Analysis

The transcriptomic analysis showed that DEGs in pathways of the plant hormone signal transduction, starch, and sucrose metabolism, phenylpropanoid biosynthesis, glycolysis/gluconeogenesis, and carbon fixation in photosynthetic organizations, and biosynthesis of amino acids of tea plants were significantly enriched in COS treatment. Metabolomics data indicated a total of 1339 differential metabolites were enriched in different metabolic pathways, including the most enriched biosynthesis of secondary metabolites and phenylpropanoid biosynthesis. A combined metabolic and transcriptomic analysis was performed to investigate the potential for the regulation of related metabolites. The result showed that the DEGs and DMs were significantly up-regulated in phenylpropane metabolism pathways. It may refer that the differential genes and metabolites in phenylpropane metabolism play an important role in the growth and development of tea plants. The DEGs and DMs in the phenylpropanoid biosynthesis pathway were summarized and sorted through the KEGG database (Appendix A). The metabolic intermediates and end products of the phenylpropanoid biosynthesis pathway were mapped to the known KEGG pathway. The overall trends of the phenylpropanoid biosynthesis pathway suggested an increase both in metabolic intermediates (such as ferulic acid and coniferyl-aldehyde) and end products (such as coniferin and sinapyl alcohol) by spraying COS on tea plants (Figure 9).

## 3. Discussion

In the past few decades, numerous studies have confirmed that chitosan oligosaccharides play an important role in promoting plant growth and development [16,21,22,23] and plant defense elicitation [13] in various crops. This present study was designed to provide more information and expand the understanding of how this elicitor COS works on tea plants, as in research [27,28] we have focused on in the past few years. We explored the function of COS on tea plants by integrating analysis of transcriptomics and metabolomics besides the yield indexes and some physiological parameters. The bud density, 100-bud-weight, and actual yield in the COS treatment group were significantly higher than those of the control group. The contents of total amino acids, chlorophyll, and soluble sugar in the tea plant were ascertainably improved t by 17.76%, 39.86%, and 64%, respectively. The actual yield (biomass) of tea was increased by 23.8%. These results concluded that COS could indubitably and significantly promote tea plant growth and yield. The chlorophyll content was remarkably increased in the COS-treated tea plant. It extensively corroborated the previous reports on tea plants [27] and other crops [16,17,35].

The chlorophyll content is an important index to measure the rate of plant photosynthesis [36]. COS could improve the content of photosynthetic pigments in various plants [37,38]. COS could induce the synthesis of phytohormones, such as gibberellins and auxin, to enhance growth and development [39] and enhance endogenous concentration [40]. Our present RNA sequencing indicated that COS activated some important metabolic activities and cell processes in tea plants. In addition. they significantly changed the expression level of genes involved in plant hormone signal transduction, photosynthesis, carbon metabolism, and amino acid metabolism. AUX1, GH3, and SAUR are the important genes encoding auxin, and the AHP gene which encodes cytokinin. Additionally, the DELLA protein (DELLA) and phytochrome interacting factor 3 (PIF3) encoding gibberellin were significantly upregulated. The above results show that spraying COS on tea plants can regulate plant growth by regulating the expression of genes related to plant hormones.

The synthesis of carbon compounds (activated by carbonic anhydrase) is directly related to photosynthesis and the growth and development of plants [41]. In the present research, COS upregulated photosynthetic organisms, starch and sucrose metabolism, and Glycolysis/Gluconeogenesis, as important components of plant carbon metabolism, played an important role in plant growth and development. The overexpressing of cytosolic FBP and SPS both resulted in the accumulation of sucrose, as reported [42,43]. Sucrose synthase (SUS) is a class of enzymes that are widely considered to be the main pathway for sucrose carbon to enter plant cell metabolism and play a key role in plant development. In our research, the chlorophyll content in the COS treatment group was increased and the genes encoding CA, FBP, SPS, and SUS were also significantly upregulated (Appendix A). The above results indicated that COS application on tea plants could enhance the photosynthetic CO_2_ fixation and the accumulation of photosynthetic assimilates, which as an endogenous correlation, further contributes to the improvement of the photosynthesis and growth of tea plants. Similarly, the content of soluble sugar in the COS treatment group was significantly higher than that in the control group.

Apart from the enhanced photosynthesis, the key genes involved in carbon and nitrogen metabolism were also upregulated by COS. It has been well-documented that carbon and nitrogen are both primary nutrients for plant growth and crop yields [44]. In the present study, genes encoding trehalose-phosphatase, PEPC, MDH, and glutamate dehydrogenase were significantly upregulated (Appendix A), which was consistent with the results obtained in rice plants [45]. Studies have proved that the net accumulation of organic acids could act as a C-skeleton for the synthesis of amino acids in nitrogen assimilation [46]. In our study, amino acid metabolism was significantly activated, and key enzymes in related metabolic pathways were significantly upregulated, such as GLUL, trpB, GS, and GDH. Metabolomics data also showed up-regulated expression of amino acid compounds, such as L-Tyrosine, L-Histidine, L-Asparagine, and L-Ornithine (Appendix A). The amount of total amino acids extracted from the tea leaves increased correspondingly in the COS-treated group. DEGs upregulated in the amino acid pathway together with the association differential metabolites of amino acid are tea-specific responses for COS treatment. In brief, COS could appreciably enhance the gene expression concerned with photosynthesis, carbon metabolism, nitrogen, amino acid metabolism, and metabolic response modification in tea plants, especially secondary metabolism, hence promoting the growth of tea plants. These related genes are the potential targets of chitosan-mediated growth stimulation in various plants [44,47,48].

Altogether, in this study, the transcriptome and metabolome of tea leaves from the control and COS treatment were performed by association analyses. Information about the differential expressed genes and metabolites was obtained. Furthermore, the transcriptome datasets and analysis reported here will facilitate functional genomics, gene discovery, and transcriptional regulation of COS in tea plants. It provided a systemic insight into the mechanism of chitosan oligosaccharides on tea plant growth promotion and formed a theoretical basis for the application of chitosan oligosaccharides in tea plants.

## 4. Materials and Methods

### 4.1. Field Experiment, Sample Collection and Yield Estimate

Field tests were conducted at a tea garden in Meitan (Longitude: 107.48 latitude: 27.7) in Guizhou Province, China. The tested tea plants (*Camellia sinensis* var. Sinensis, Qianmei 601.) were around 10 years old and free of pesticide application during the test.

5% COS agents were purchased from Hainan Zhengye Zhongnong High-tech Co., Ltd., Haikou, China. The commercial agent was diluted with water 500, 800, and 1000 times in volume, and water consumption was 32 kg for one mu (667 m^2^). The actual dose of COS was 3.2, 2.0, and 1.6 g/667 m^2^, respectively. The adjacent plot with no COS spraying served as a control (CK). There were three replicates in the randomized complete block for every spraying application in the tea garden. Seven days after spraying COS, yields in the control and COS-treated group were allowed to estimate in randomly selected tea plant rows in the center of the field. The number of buds in each of 0.1 m^2^ (0.33 m × 0.33 m) was counted, and the mean bud density (buds per square meter) for each plot was calculated. The obtained tea buds were randomly selected, and the weight of the 100 tea buds was measured. All measurements were repeated three times. Finally, the tea buds picked in each treatment group were quickly frozen with liquid nitrogen and stored in a −80 °C freezer.

### 4.2. Determination of Chlorophyll, Soluble Sugar and Amino Acid Content

In the controls and COS-treated leaves, the content of chlorophyll soluble sugar and amino acids was measured and expressed as mg/g FW (fresh weight). Among them, chlorophyll and soluble sugar content were determined by following the provisions in bioassay kits, as previously reported [28]. The amino acid content was determined according to GB/T8314-2013 determination of total free amino acids in tea. The involved bioassay kits were purchased from Solarbio, Beijing, China.

### 4.3. Transcriptomic Profiling

RNA sequencing was used to analyze the transcriptome of samples from the control and COS treatment groups, treated at a concentration of 2.0 g/667 m^2^ (*n* = 3 for both groups, six samples in total) as previous description [28]. The extraction of total RNAs was done by using TRIzol (Invitrogen, Carlsbad, CA, USA). The concentration of isolated RNA was determined by NanoDrop TM OneC spectrophotometer (Thermo Fisher Scientific Inc., Carlsbad, CA, USA). The quality and integrity of RNA were checked by running a 1.5% agarose gel. The cDNA synthesis was finished by using 2 ug DNAse-treated RNA using a cDNA synthesis kit (Thermo Fisher Scientific Inc., Carlsbad, CA, USA). The mRNA libraries were generated using KC-Digital^TM^ stranded mRNA libraryprep kit for Illumina^®^ (Catalog NO. DR08502, Seqhealth Technology Co., Ltd., Wuhan, China), and the library quality was assessed on the Agilent Bioanalyzer 2100 system. The kit eliminates duplication bias in PCR and sequencing steps by using a unique molecular identifier (UMI) of eight random bases to label the pre-amplified cDNA molecules. The library products corresponding to 200–500 bps were enriched, quantified, and finally sequenced on Illumina Novaseq 6000 (Illumina, San Diego, CA, USA). RNA extraction, library preparation, and data analysis of high-throughput sequencing were completed by Kangce Technology Co., Ltd. in Wuhan, China. For RNA-Seq data analysis, raw sequencing data were first analyzed by FastQC, and low-quality reads were filtered by Trimmomatic (version 0.36) and discarded. The reads contaminated with adaptor sequences were trimmed, and then clean reads were mapped to the reference genome of *Camellia sinensis* using STATR software (version 2.5.3a). Estimation of gene expression level using Fragments Per Kilobase Million (FPKM) method. All assembled unigenes were annotated in the following databases: GO and KEGG. Differentially expressed genes (DEGs) between the two groups were identified using edgeR software (version 3.12.1). FDR-corrected *p* < 0.05 with a fold change ≥ 2 was used to determine whether the differential genes were statistically significant. Differential gene GO, and KEGG data analysis was performed using KOBAS software (version 2.1.1) with a corrected *p* < 0.05 to determine if the enrichment was statistically significant. To infer the putative functions of DEGs, we conducted GO enrichment analysis using top GO and KEGG pathways and enrichment analysis using cluster Profiler(version R 3.6.0, October 2019).

### 4.4. Metabolite Profiling

#### 4.4.1. Extraction and Detection of Metabolites

Tea leaves (100 mg) from the COS treatment (HAI7_2) and control (KB7_1) were ground with liquid nitrogen, and the homogenate was resuspended with prechilled 80% methanol and 0.1% formic acid using a vortex. The samples were incubated on ice for 5 min and then centrifuged at 15,000× *g*, 4 °C for 20 min. Some supernatants were diluted to a final concentration containing 53% methanol with LC-MS-grade water. The samples were subsequently transferred to a fresh Eppendorf tube and were then centrifuged at 15,000× *g*, 4 °C for 20 min. Finally, the supernatant was injected into the LC-MS/MS system analysis [49]. UHPLC-MS/MS analyses were performed using a Vanquish UHPLC system (Thermo Fisher Scientific GmbH, Dreieich, Germany) coupled with an Orbitrap Q Exactive^TM^ HF-X mass spectrometer (Thermo Fisher, Dreieich, Germany) by Novogene Co., Ltd. in Beijing, China. The raw data files generated by UHPLC-MS/MS were processed using Compound Discoverer 3.1 (Thermo Fisher, Dreieich, Germany) to perform peak alignment, peak picking, and quantitation for each metabolite. Peaks were then matched with the mzCloud (https://www.mzcloud.org/, accessed on 26 October 2020), mzVault, and MassList databases to obtain accurate qualitative and relative quantitative results. Statistical analyses were performed using the statistical software R (R version R-3.4.3), Python (Python 2.7.6 version), and CentOS (CentOS release 6.6); when data were not normally distributed, normal transformations were attempted using of area normalization method. The metabolomic sampling method is the same as the transcriptome analysis experiment, and three biological replicates are set for each treatment.

#### 4.4.2. Data Processing and Statistics

These metabolites were annotated using the KEGG database (https://www.genome.jp/kegg/pathway.html, accessed on 26 Otcober 2020), HMDB database (https://hmdb.ca/metabolites, accessed on 26 Otcober 2020), and LIPI Maps database (http://www.lipidmaps.org/, accessed on 26 Otcober 2020). Principal components analysis (PCA) was performed at meta-X (a flexible and comprehensive software for processing metabolomics data). We applied univariate analysis (*t*-test) to calculate the statistical significance (*p*-value). The metabolites with VIP > 1 and *p*-value < 0.05 and fold change(FC) ≥ 2 or FC ≤ 0.5 were considered to be differential metabolites.

For clustering heat maps, the data were normalized using z-scores of the intensity areas of differential metabolites and were plotted by pheatmap package in R language. The correlation between differential metabolites was analyzed by “cor ()” in R language (method = “Pearson”). Statistically significant correlation between differential metabolites was calculated by “cor. test ()” in R language. A *p*-value < 0.05 was considered statistically significant, and correlation plots were plotted by corrplot package in R language. The functions of these metabolites and metabolic pathways were studied using the KEGG database.

### 4.5. Validation of qRT-PCR

To verify the reliability of transcriptome sequencing results, we selected seven DEGs associated with tea plant defense for qRT-PCR analysis. Gene-specific primers (Appendix A) were designed. Additionally, qRT-PCR was performed on a Light Cycler 96 System in a 20 mL reaction volume under the following parameters: 95 °C for 15 s, 60 °C for 30 s, and 72 °C for 30 s for 40 cycles. GAPDH was used as a reference gene, and relative gene expression was calculated using the 2^−ΔΔCt^ method [50].

### 4.6. Statistical Analysis

Data were expressed as the mean ± standard error, and the data were subjected to a one-way analysis of variance (ANOVA A) (*p* < 0.05) followed by a significant difference test (LSD) using SPSS statistics v16.0 (SPSS Inc., Chicago, IL, USA).

## Figures and Tables

**Figure 1 ijms-23-05469-f001:**
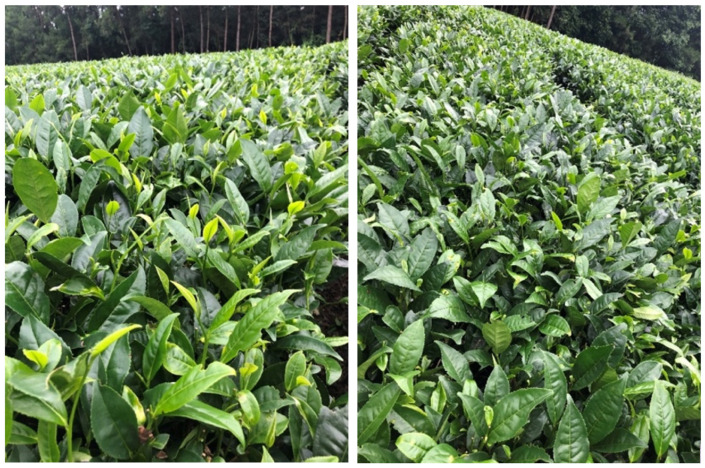
Tea plants bearing fresh tea shoots seven days after treatment (**left**: COS-treated tea plant at the concentration of 2.0 g/667 m^2^; **right**: CK).

**Figure 2 ijms-23-05469-f002:**
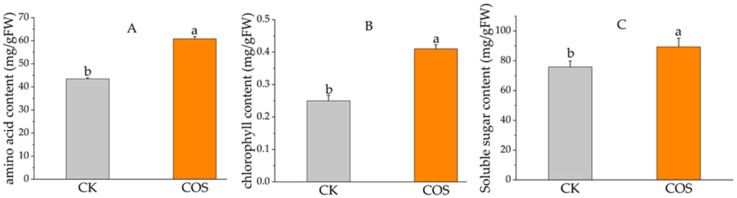
Effects of COS on amino acid, chlorophyll, and soluble sugar content. (**A**) Amino acid content; (**B**) Chlorophyll content; (**C**) Soluble sugar content. The value of the vertical axis represents the mean ± SD of three biological repeats. Different letters indicate significant differences at *p* < 0.05.

**Figure 3 ijms-23-05469-f003:**
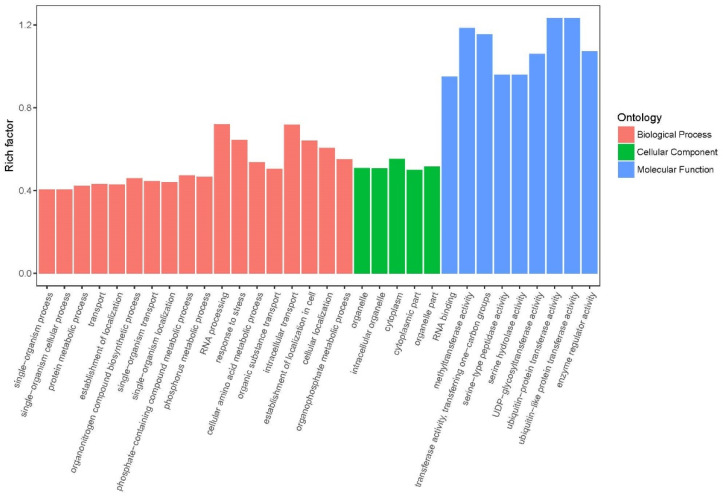
GO classification analysis based on DEGs.

**Figure 4 ijms-23-05469-f004:**
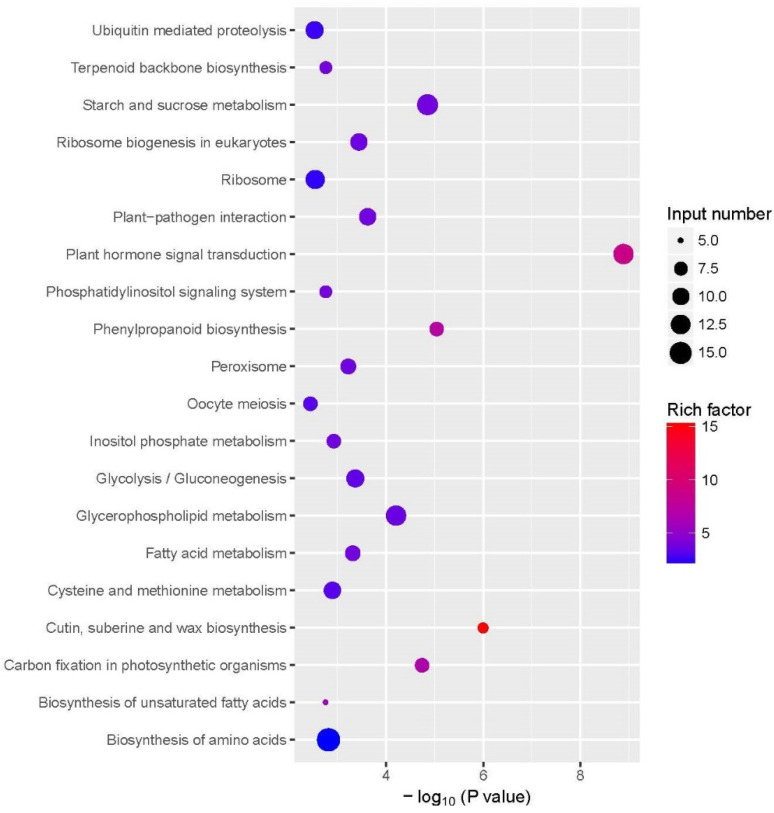
KEGG enrichment analysis of upregulated DEGs response to COS treatment.

**Figure 5 ijms-23-05469-f005:**
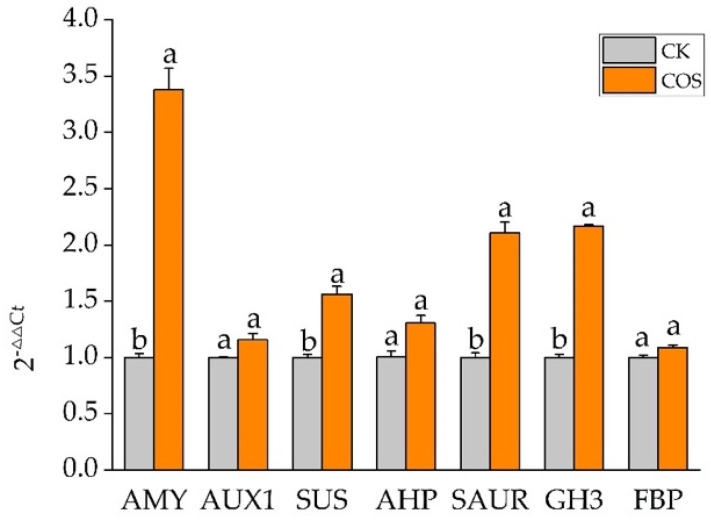
qRT-PCR analysis on randomly selected DEGs.

**Figure 6 ijms-23-05469-f006:**
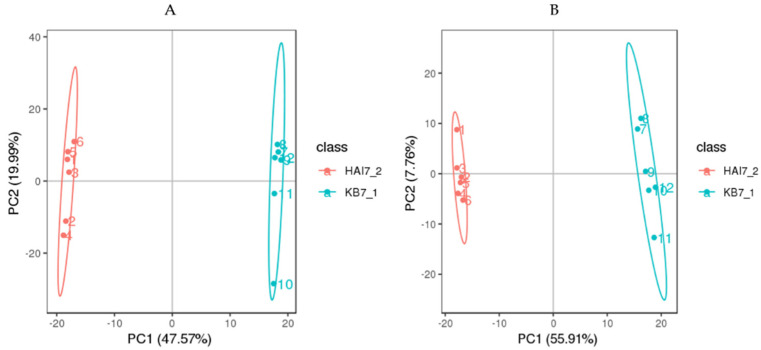
Principal components analysis (PCA) of metabolites. (**A**) is the positive ion mode, and (**B**) is the negative ion mode. Red represents the samples of COS treatment (HAI7_2), and the green represents control (KB7_1).

**Figure 7 ijms-23-05469-f007:**
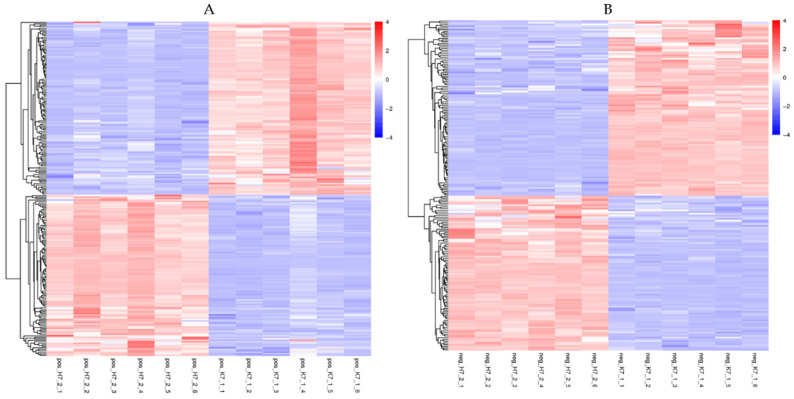
Heatmap of metabolites. Red indicates high abundance, whereas low relative metabolites are shown in blue ((**A**): positive ion mode, (**B**): negative ion mode). COS treatment sample: H7_2_1-H7_2_6; control: K7_1_1-K7_1_6).

**Figure 8 ijms-23-05469-f008:**
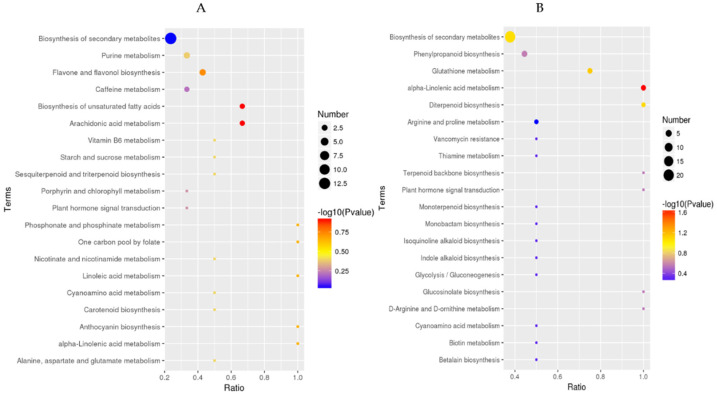
KEGG analysis at positive ion mode (**A**) and negative ion mode (**B**). The abscissa is x/y (x: number of differential metabolites in the corresponding metabolic pathway, y: total metabolites identified in this pathway). The higher the value, the higher the enrichment of differential metabolites in this pathway.

**Figure 9 ijms-23-05469-f009:**
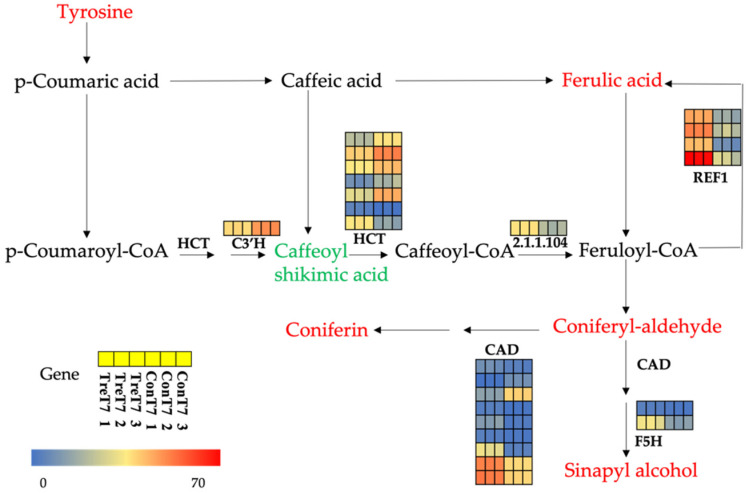
Schematic of a portion of phenylpropanoid biosynthesis pathway in tee plant. Among them, the differential metabolites are marked in red and green, red means up-regulated, and green means down-regulated. Differentially expressed genes are marked next to related enzymes in the form of a heat map. The gene scale bar corresponds to the range of relative transcriptional levels of enzymes.

**Table 1 ijms-23-05469-t001:** Tea yield compared between CK and COS treatment in field test.

Yield Indicators	CK	COS Treatment (with Different Times of Dilution)		
		3.2 g/667 m^2^ (1:500)	2.0 g/667 m^2^ (1:800)	1.6 g/667 m^2^ (1:1000)
Bud density (bud/m^2^)	1028 ± 48b	1223 ± 60a	1288 ± 38a	1269 ± 64a
100-bud weight (g/100 bud)	6.76 ± 0.04b	7.68 ± 0.07a	7.77 ± 0.04a	7.70 ± 0.07a
Actual output (g/m^2^)	29.35 ± 0.10b	33.96 ± 0.04a	36.33 ± 0.06a	34.66 ± 0.08a

The value represents the mean ± SD of three biological repeats. Different letters indicate significant differences at *p* < 0.05.

**Table 2 ijms-23-05469-t002:** Throughput and quality summary of RNA-sequence.

Samples	Raw Reads	Clean Reads	Q20 (%)	Q30 (%)	GC (%)
TreT7_1(COS)	84011624	78935806	100	99.4	46.81
TreT7_2(COS)	106371756	99995940	100	99.3	46.73
TreT7_3(COS)	70432706	66250686	100	99.3	46.73
ConT7_1(CK)	81114666	75623100	100	99.3	46.94
ConT7_2(CK)	96205136	89960092	100	99.3	46.74
ConT7_3(CK)	76155870	70900816	100	99.3	46.8

**Table 3 ijms-23-05469-t003:** Metabolites test results.

Compared Samples	Total Metabolites	Total DMs	Upregulated DMs	Down Regulated DMs
COS vs. control pos	748	224	108	116
COS vs. control neg	591	192	90	102

## Data Availability

All data in the manuscript are available from the corresponding author upon request.

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
