# Peer review of "Transcriptomic and Metabolomic Analysis Reveal Possible Molecular Mechanisms Regulating Tea Plant Growth Elicited by Chitosan Oligosaccharide"

_ijms, 2022, doi:10.3390/ijms23105469_

Round 1
Reviewer 1 Report
The manuscript describes the multi-omics analyses to elucidate the positive effect of chitosan oligosaccharide on tea plant growth, and suggests the addition of chitosan promotes plant growth due to improved primary metabolism.
The manuscript showed the results of two omics data (transcriptome and metabolome) individually. The first half of the paper describes the similar analysis as in the authors' previous study (Ou et al. 2022). It could be worth showing integrated multi-omics analyses ex. activated pathway with both of up/down-regulated gene expression and metabolites like Pang (2022) et al. and Wang et al.(2020).
It is worth to show the tea-specific results compared to the other transcriptome or metabolome studies of chitosan effect on plants (potato by Lemke et al. 2020, strawberry by Landi et al. 2017).
L121 Closing parenthesis would be necessary after "COS".
L344 Period after "group" should be removed.
L348 Methods of transcriptome assembly is not described - please add description of assembling pipeline (de novo or genome-guided).
L349 It may be appropriate to include references and version information regarding software and R-packages (ex. KOBAS, edgeR, topGO).
Author Response
Response to reviewer 1
The manuscript describes the multi-omics analyses to elucidate the positive effect of chitosan oligosaccharide on tea plant growth, and suggests the addition of chitosan promotes plant growth due to improved primary metabolism. The manuscript showed the results of two omics data (transcriptome and metabolome) individually. The first half of the paper describes the similar analysis as in the authors' previous study (Ou et al. 2022). It could be worth showing integrated multi-omics analyses ex. activated pathway with both of up/down-regulated gene expression and metabolites like Pang (2022) et al. and Wang et al.(2020).
Dear esteemed reviewer,
Thank you so much for the helpful comments and suggestions. We make meticulous investigation and response to your comments as listed below.
- It is worth to show the tea-specific results compared to the other transcriptome or metabolome studies of chitosan effect on plants (potato by Lemke et al. 2020, strawberry by Landi et al. 2017).
Responses: Thanks for your suggestion. And we add this part of the description as follows: "Among them, glutamine synthetase (GLUL), and tryptophan(trpB) synthase are DEGs specifically belonging to tea in biosynthesis amino acids." in section 2.4.2 and " In our study, amino acid metabolism was significantly activated, and key enzymes in related metabolic pathways were significantly up regulated, such as GLUL, trpB, GS, and GDH. Metabolomics data also showed up-regulated expression of amino acid compounds, such as L-Tyrosine, L-Histidine, L-Asparagine, and L-Ornithine (TableS4) which s. The amount of total amino acids extracted from tea leaves increased correspondingly in the COS-treated group. DEGs up regulated in the amino acid pathway together with the association differential metabolites of amino acid are tea-specific response for COS treatment."
- L121 Closing parenthesis would be necessary after "COS".
Responses: Done.
- L344 Period after "group" should be removed.
Responses: Done.
- L348 Methods of transcriptome assembly is not described - please add the description of assembling pipeline (de novo or genome-guided).
Responses: Thanks. We have added a description to the methods of transcriptome assembly in that section.
- L349 It may be appropriate to include references and version information regarding software and R-packages (ex. KOBAS, edgeR, topGO).
Responses: Done. “KOBAS software (version 2.1.1)” and “edgeR software” (version 3.12.1) were included. The “topGO” indicated that the enrichment was the most abundant GO cluster, which was the result of our screening.

Reviewer 2 Report
The review of the paper written by Ji et al.
The paper described the effect of chitosan application to leave in tea plants grown under field condition. I do not understand, why Chitosan is written with upper case.
There is no information on Figure 1 about the variant of COS. What COS treatment is presented here?
The order on Figure 2 is strange. In the figure caption there is: chlorophyll, soluble sugar, and amino acids, but then the order is amino acids, chlorophyll, and soluble sugars. I do not understand why it is not the same.
The discussion is quite poorly written. There is lack of link to other papers with similar results, or results, which can partially confirm your statements. There are also some sentences, which should be change. In the discussion very few papers is cited. Moreover most of them are older than 5 years. Some newer papers were cited in the Introduction section.
L232 “Auxin is the first plant hormone discovered by humans…” who else than humans could discovered auxin?
Such sentences like in L240-1 and 245-6 need to be confirmed by some analysis to verify it. You have not measure the auxin accumulation as well as ABA accumulation nor cite results paper confirming such link.
The conclusion is in my opinion over speculative.
The metabolomic and transcriptomic analysis are presented very generally. In my opinion more information about results could be described. You only point out the most important metabolic processes, which are affected. Some of the results are mentioned in the discussion section. You should referred in the discussion section only to this results which are presented in the result section.
In my opinion the results and discussion should be rewrite or even write from the beginning.
Author Response
Response to reviewer 2
Dear esteemed reviewer,
Thank you so much for the helpful comments and suggestions. We make meticulous investigation and response to your comments as listed below.
- The paper described the effect of chitosan application to leave in tea plants grown under field condition. I do not understand, why Chitosan is written with upper case.
Responses: Thanks for your valuable comments and suggestions. We've changed it to lowercase "chitosan".
- There is no information on Figure 1 about the variant of COS. What COS treatment is presented here?
Responses: We have added "2.0g/667m2" under Figure 1, which was an 800 times dilution. Relevant content has also been added in the materials and methods section.
- The order on Figure 2 is strange. In the figure caption there is: chlorophyll, soluble sugar, and amino acids, but then the order is amino acids, chlorophyll, and soluble sugars. I do not understand why it is not the same.
Responses: Thanks. We have adjusted as" Effects of COS on amino acid, chlorophyll and soluble sugar content."
- The discussion is quite poorly written. There is lack of link to other papers with similar results, or results, which can partially confirm your statements. There are also some sentences, which should be change. In the discussion very few papers is cited. Moreover most of them are older than 5 years. Some newer papers were cited in the Introduction section.
Responses: Thanks for your valuable comments and suggestions. We thoroughly analyzed the results and rewrote the 'Results and Discussion” section, citing some new literature, and adding some new content.
- L232 “Auxin is the first plant hormone discovered by humans…” who else than humans could discovered auxin?
Responses: We deleted "by humans" and paragraphed this description.
- Such sentences like in L240-1 and 245-6 need to be confirmed by some analysis to verify it. You have not measure the auxin accumulation as well as ABA accumulation nor cite results paper confirming such link.
Responses: Thanks, and sure we agree with you. As suggested, when we describe conclusive sentences, we include references or are supported by our data. For example, "auxin" was mentioned in the DEGs and related references. This section of discussion was rewritten.
- The conclusion is in my opinion over speculative.
Responses: Thanks. We rewrote this part as "Altogether, in this study, the transcriptome and metabolome of tea leaves from the control and COS treatment were performed by association analyses, and information about the differential expressed genes and metabolites were obtained. And the tran-scriptome datasets and analysis reported here will facilitate functional genomics, gene discovery, and transcriptional regulation of COS on tea plant. It provides a systemic in-sight into the mechanism of chitosan oligosaccharides on tea plant growth promotion and formed a theoretical basis for the application of chitosan oligosaccharides in tea plants."
- The metabolomic and transcriptomic analysis are presented very generally. In my opinion more information about results could be described. You only point out the most important metabolic processes, which are affected. Some of the results are mentioned in the discussion section. You should referred in the discussion section only to this results which are presented in the tion.
Responses: Thanks for your valuable comments and suggestions. We have re-described the "Results and Discussion" section with new references added and data reorganized and analyzed.
- In my opinion the results and discussion should be rewrite or even write from the beginning.
Responses: Thanks. We have reworked the "Results and Discussion “section from the beginning.

Round 2
Reviewer 1 Report
I would like to recommend to show in the figure the activated pathways identified by both of the transcriptome and metabolome like Fig. 7d in Pang et al. (2022) https://doi.org/10.1186/s12864-022-08341-x.
In the previous revision, the author has added a note about the RNA-seq experiment, but it need to show the data analysis pipeline (how the transcriptome sequences were obtained from the fastq data, ex. settings for using assemble software such as Trinity or Stringtie), including where to download the raw sequencing data (SRA/ERA/DRA accession number).
Author Response
Response to reviewer
Dear esteemed reviewer,
Thank you for the helpful comments and suggestions. We make careful investigation and response to your comments as listed below.
- I would like to recommend to show in the figure the activated pathways identified by both of the transcriptome and metabolome like Fig. 7d in Pang et al. (2022) https://doi.org/10.1186/s12864-022-08341-x.
Responses:
Thanks for your suggestion. According to Pang et al. (2022), we have constructed the phenylpropanoid biosynthesis pathway identified by both transcriptome and metabolome (Figure 9). the description and combination analysis were added as follows:
“2.4. Combination of the metabolic profiling and transcriptomic analysis
The transcriptomic analysis showed that DEGs in pathways of the plant hormone signal transduction, starch and sucrose metabolism, phenylpropanoid biosynthesis, glycolysis/gluconeogenesis, and carbon fixation in photosynthetic organizations, and biosynthesis of amino acids of tea plants were significantly enriched in COS treatment. Metabolomics data indicated a total of 1339 differential metabolites were enriched in different metabolic pathways, including the most enriched biosynthesis of secondary metabolites and phenylpropanoid biosynthesis. A combined metabolic and transcriptomic analysis was performed to investigate the potential for the regulation of related metabolites. The result showed that the DEGs and DMs were significantly up-regulated in phenylpropane metabolism pathways. It may refer that the differential genes and metabolites in phenylpropane metabolism play an important role in the growth and development of tea plants. The DEGs and DMs in the phenylpropanoid biosynthesis pathway were summarized and sorted through the KEGG database (Table S12). The metabolic intermediates and end products of the phenylpropanoid biosynthesis pathway were mapped to the known KEGG pathway. The overall trends of the phenylpropanoid biosynthesis pathway suggested an increase both in metabolic intermediates (such as ferulic acid, and coniferyl-aldehyde) and end products (such as coniferin and sinapyl alcohol) by spraying COS on tea plants (Figure 9).
Figure 9. Schematic of a portion of phenylpropanoid biosynthesis pathway in tee plant. Among them, the differential metabolites are marked in red and green, red means up-regulated, and green means down-regulated. Differentially expressed genes are marked next to related enzymes in the form of heat map. Gene scale bar corresponds to the range of relative transcriptional levels of enzymes."
However, regarding our transcriptomic data, several pathways that are significantly enriched, like plant hormone signal transduction, starch and sucrose metabolism, phenylpropanoid biosynthesis, glycolysis/gluconeogenesis, and carbon fixation in photosynthetic organizations, and biosynthesis of amino acids, which are generally considered to be major contributors to the growth and development of tea plants. While in the metabolomic data, the differential metabolites were obviously enriched in "biosynthesis of secondary metabolites", only a few DMs were correlated in the growth and development pathway of tea plants. DEGs did not necessarily cause the same expression trend of differential metabolites, or the substantial DMs could not adequately be representative of the DEGs, which brought great difficulties to a crucial analysis.
Although we have constructed the phenylpropanoid biosynthesis pathway identified by both transcriptome and metabolome, it seems limited to conclude its contribution to tea plant growth and development. We suggest excluding this newly added part "2.4. Combination of the metabolic profiling and transcriptomic analysis" in the main text and listed as the expanding content in the supporting material.
- In the previous revision, the author has added a note about the RNA-seq experiment, but it need to show the data analysis pipeline (how the transcriptome sequences were obtained from the fastq data, ex. settings for using assemble software such as Trinity or Stringtie), including where to download the raw sequencing data (SRA/ERA/DRA accession number).
Responses: Thanks for your suggestion. In the "Materials and Methods" section, we have added related information about the acquisition method of RNA-seq, analysis software, and analysis process.
